# Classification of virulence factors based on dual-channel neural networks with pre-trained language models

Guanghui Li[1]☉*, Peiyang Song[1]☉, Jiawei Luo[2], Cheng Liang[3]*

1 School of Information and Software Engineering, East China Jiaotong University, Nanchang, China,
2 College of Computer Science and Electronic Engineering, Hunan University, Changsha, China,
3 School of Information Science and Engineering, Shandong Normal University, Jinan, China

☉ These authors contributed equally to this work.
* ghli16@hnu.edu.cn (GL); alcs417@sdnu.edu.cn (CL)

## Abstract

Virulence factors (VFs) are crucial molecules that enable pathogens to cause infection and disease in a host. They allow pathogens to evade the host's immune defenses and facilitate the progression of infection through various mechanisms. With the increasing prevalence of antibiotic-resistant strains and the emergence of new and re-emerging infectious agents, the classification of VFs has become more critical. This study presents PLM-GNN, an innovative dual-channel model designed for precise classification of VFs, focusing on the seven most numerous types. It integrates a structure channel, which employs a geometric graph neural network to capture the three-dimensional structure features of VFs, and a sequence channel that utilizes a pre-trained language model with Convolutional Neural Network (CNN) and Transformer architectures to extract local and global features from VF sequences, respectively. On the independent test set, the method achieved an accuracy of 86.47%, an F1 score of 86.20% and an Area Under the Receiver Operating Characteristic Curve (AUC) of 97.20%, validating its effectiveness. In conclusion, PLM-GNN can precisely classify the seven major VFs, offering a novel approach for studying their functions.

## Introduction

Infectious diseases continue to represent a growing and persistent threat to human health, placing considerable strain on global public health systems [1]. Bacterial infections are primarily mediated by VFs, which facilitate pathogens in initiating and maintaining infections. The horizontal gene transfer capacity of VFs across distinct bacterial strains or species significantly elevates the likelihood of novel pathotype emergence, rendering these occurrences nearly unavoidable [2]. Therefore, identifying the VFs of pathogenic bacteria is essential for understanding the mechanisms

**Data availability statement:** The source code and associated datasets for this study are publicly available on GitHub at: https://github.com/ghli16/PLM-GNN.

**Funding:** This work was supported by the National Natural Science Foundation of China [grant numbers 62362034, 62372279]; the Natural Science Foundation of Jiangxi Province of China [grant number 20232ACB202010]; the Natural Science Foundation of Shandong Province [grant number ZR2023MF119]; and the Jiangxi Province Key Laboratory of Advanced Network Computing [grant number 2024SSY03071]. The funders had no role in study design, data collection and analysis, decision to publish, or preparation of the manuscript.

**Competing interests:** The authors declare no competing financial interest.

of pathogenesis and for discovering potential targets for novel drug development and vaccine design, thereby enhancing our capacity to prevent and treat infectious diseases [3].

Moreover, the diversity and complexity of VFs necessitate a multifaceted classification system to effectively categorize and understand these elements. VFs encompass diverse categories, comprising secreted proteins (e.g., *protein toxins* and *enzymes*) as well as cell-surface components such as *capsular polysaccharides*, *lipopolysaccharides*, and *outer membrane proteins* [4]. A systematic categorization of VFs not only enhances the comprehension of molecular mechanisms driving bacterial pathogenesis but also supports the discovery of innovative therapeutic interventions and potential vaccine development [5].

The establishment of comprehensive VF databases, such as VFDB [6], Victors [7], and MvirDB [8], has made the identification of VFs increasingly feasible. A predominant strategy currently employed for VF prediction relies on sequence similarity to known VFs. For example, Liu et al. created VRprofile [9], a tool capable of identifying homologs of conserved gene clusters through Hidden Markov Model ER (HMMer) [10] or Basic Local Alignment Search Tool for Proteins (BLASTp) searches. This enables the prediction of virulence factors and antibiotic resistance genes within the genomes of pathogenic bacteria. Furthermore, Liu et al. [2] developed the VFanalyzer platform, an online tool that employs a hybrid methodology integrating BLAST [11] and hidden Markov models (HMMs) to perform iterative sequence similarity analyses within the VFDB. This system is designed to improve the precision of VF identification in bacterial genomes. Despite the widespread adoption of sequence alignment-based approaches for VF prediction, these methods exhibit significant limitations. Specifically, their reliance on sequence similarity restricts their ability to detect conserved VFs, while novel VFs with divergent evolutionary origins often remain undetected.

To overcome these challenges, recent studies have adopted machine learning and deep learning frameworks. Garg et al. [12] introduced VirulentPred, a prediction method designed to identify bacterial virulence proteins using a two-layer cascade Support Vector Machine (SVM). In this approach, the initial layer of classifiers is trained on a variety of sequence features and Position-Specific Scoring Matrices (PSSM). The outputs from these classifiers are then fed into a second-layer SVM for additional training and final prediction. Gupta et al. [13] developed MP3, a standalone tool and web server that employs an integrated approach combining SVM and Hidden Markov Models to perform rapid, sensitive, and accurate predictions of pathogenic proteins. Rentzsch et al. [14] proposed a novel strategy for selecting negative samples to construct the dataset, combining sequence similarity with machine learning models, which yielded promising results. DeepVF [15] thoroughly explored a diverse set of heterogeneous features using well-established machine learning algorithms. They utilized four traditional machine learning algorithms and three deep learning methods to train 62 baseline models. These models' strengths were then effectively integrated, and a stacking strategy was applied to identify the VFs. Singh et al. [16] proposed VF-Pred, which integrates a novel sequence alignment-based

feature (SeqAlignment) to significantly enhance the accuracy of machine learning-driven predictions. The model leveraged 982 features derived from rigorous feature engineering and implemented a downstream ensemble strategy, integrating outputs from 25 distinct models to enhance VF identification. In the classification of VFs, Zheng et al. [17] constructed a comprehensive dataset containing 160,495 virulence protein sequences categorized into 3,446 classes. Based on this dataset, a neural network architecture was developed, comprising two convolutional layers alternating with two max-pooling layers and a subsequent multilayer perceptron (MLP) for VF classification.

With advancements in technology, several pre-trained language models [18–20] for proteins have emerged. These models were trained on large datasets using stacked Transformer blocks, ultimately generating embeddings for each amino acid. Previous studies have indicated that protein representations extracted from pre-trained language models achieve state-of-the-art performance across multiple tasks [21,22]. Sun et al. [23] developed DTVF, which utilized features generated by ProtT5 and integrated two channels—Convolutional Neural Networks and Long Short-Term Memory (LSTM). By incorporating an attention mechanism, this approach significantly enhanced the accuracy of VF recognition. However, the aforementioned model primarily emphasizes the sequence information, neglecting the Three Dimension (3D) structure of the VFs.

The sequence-structure-function paradigm posits that the amino acid sequence dictates the protein's spatial configuration, which in turn governs its function [24]. Consequently, the 3D structure plays a crucial role in classifying VFs. Recently, ESMFold [19] developed by Meta AI has demonstrated the ability to generate 3D structures directly from protein sequences, achieving accuracy comparable to AlphaFold2 [25], but with greater speed and efficiency. Unlike AlphaFold2, which depends on multiple sequence alignment (MSA) data for its predictions, ESMFold can accurately predict 3D structures without requiring additional MSA information. This capability provides a significant advantage in scenarios where obtaining sufficient homologous sequences is challenging. This capability opens new avenues for using structure approaches to predict VFs. For example, GTAE-VF [26] leveraged 3D structures predicted by ESMFold to transform the VF recognition challenge into a graph-level prediction task. As a Graph Transformer Autoencoder, GTAE-VF integrated the strengths of Graph Convolutional Network (GCN [27]) and the Transformer [28] architecture, enabling full pairwise message passing. This allowed GTAE-VF to learn both local and global information adaptively, thereby capturing long-range dependencies and latent patterns more effectively. GTAE-VF achieved robust and reliable prediction accuracy, validating the utility of using 3D structures for VF identification. This approach demonstrates that integrating advanced structural predictions with graph-based ML models can significantly enhance our understanding and predictive power in microbial pathogenesis.

Previous approaches exhibit notable limitations: 1) The majority of existing models are confined to binary classification of VFs (e.g., classifying them as "VFs" or "non-VFs"), with limited exploration of multi-class classification for VFs. 2) Moreover, current mainstream models depend exclusively on either sequence-based or structure-based methods, neglecting the integration of both sequence and structure information into a cohesive framework. 3) Currently, models for identifying VFs that utilize structure information rely solely on the distances between alpha carbon atoms to construct contact graphs, failing to fully exploit the geometric information available [29].

Therefore, in this study, we propose the PLM-GNN, a dual-channel framework for classifying VFs by integrating **p**re-trained **l**anguage **m**odels and **g**eometric graph **n**eural **n**etworks. In the sequence channel, ESM-2 is employed to extract 1280-dimensional feature representations for each amino acid, capturing complex sequence patterns, deep semantic information, and evolutionary relationships. These features are then processed through a two-layer one-dimensional convolutional neural network [30] to derive local sequence features, followed by a Transformer encoder to extract global sequence dependencies. In the structure channel, ESMFold is utilized to predict the three-dimensional structure information of VFs, while ProtT5 enhances node-level representations. These structure features are subsequently fed into a geometric graph neural network for further processing. Finally, the sequence and structure embeddings are fused and passed through an MLP [31] for classification. Extensive experimental results demonstrate that PLM-GNN effectively captures

discriminative features of various VF categories and achieves state-of-the-art performance in multi-class classification tasks.

Our key contributions are summarized as follows:

- Existing models for VF identification and classification rely exclusively on either sequence or structure information, failing to leverage the complementary nature of both data types. Our approach addresses this limitation by introducing a dual-channel framework that integrates sequence and structure data.

- We propose PLM-GNN, a novel dual-channel model for VF classification, which achieves superior performance by effectively fusing sequence and structure features.

- Comprehensive experiments on multiple datasets demonstrate that PLM-GNN not only excels in VF classification but also generalizes well to other protein classification tasks, highlighting its versatility and robustness.

## Methods

### Dataset description

The VFDB core database comprises 14 categories of VFs, totaling 4,236 sequences. Among these, the seven least-represented categories collectively contain only 216 sequences (averaging just 30 per category). Given this limited data, it is likely that deep learning models would struggle to learn adequate features from such sparse categories. Therefore, we selected the seven most abundant VF categories, comprising a total of 4,020 sequences, to constitute our initial dataset.

To enhance the generalization capacity and computational efficiency of the dataset, CD-HIT [32] was employed to remove VFs sharing sequence homology exceeding 90%. Due to the current GPU memory capacity being limited to 24 GB, the ESMFold model can only predict three-dimensional structures of proteins with amino acid sequences no longer than 1,240 residues. Therefore, we further removed sequences longer than 1,240 amino acids, which amounted to 128 entries, accounting for 3.29% of the resulting dataset. The length distribution of the resulting dataset, as shown in Fig 1, indicates that long VFs constitute only a minimal proportion. Additionally, we analyzed the length distribution for each VF type, as illustrated in Fig 2. The results reveal that very long VFs (exceeding 1240 amino acid residues) account for a

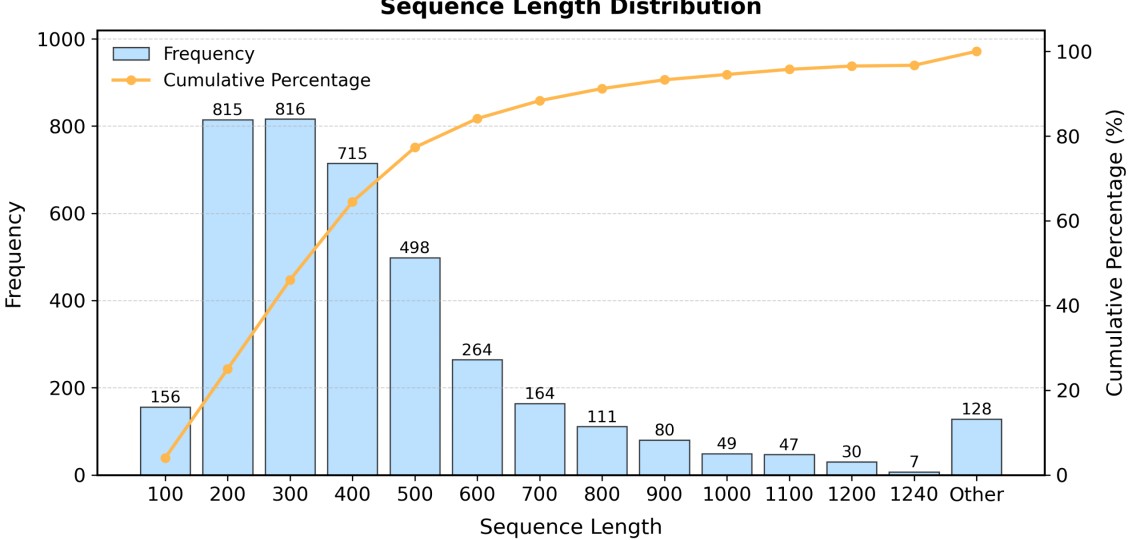

**Fig 1. The distribution of sequence lengths, including frequency distribution and cumulative percentage distribution.**

**Fig 2. The length distribution of various types of VFs.**

relatively small proportion of each class. Based on these findings, we reasoned that the data processing strategy would have minimal impact on the overall prediction performance. To ensure that the class distribution—particularly that of minority classes—in the training, validation, and test sets remained consistent with the original dataset, we applied stratified sampling. This method effectively prevents issues such as underrepresentation or complete absence of minority class samples in any subset, which might otherwise arise from random partitioning. For the data splitting ratios, we compared three different configurations: 7.5:1.5:1.5, 8:1:1, and 9:0.5:0.5. Based on the results shown in Fig 2, the model performed best under the 8:1:1 ratio, which was therefore selected for stratified sampling. This strategy ensures that each subset is representative and reliable, thereby supporting robust model training and enabling unbiased performance evaluation. The resulting data distribution is presented in Table 1.

**Table 1. Statistics on the number of VF classes and the distribution of the dataset.**

| classes | ID | Train set | Valid set | Test set | Total |
|---|---|---|---|---|---|
| Adherence | VFC0001 | 438 | 55 | 55 | 548 |
| Effector delivery system | VFC0086 | 1264 | 158 | 158 | 1580 |
| Motility | VFC0204 | 406 | 51 | 51 | 508 |
| Exotoxin | VFC0235 | 156 | 20 | 20 | 196 |
| Immune modulation | VFC0258 | 471 | 59 | 59 | 589 |
| Biofilm | VFC0271 | 67 | 8 | 9 | 84 |
| Nutritional/Metabolic factor | VFC0272 | 197 | 25 | 25 | 247 |

### An overview of PLM-GNN

The overall workflow of PLM-GNN is illustrated in Fig 3, which consists of four main parts: (1) obtaining the initial sequence feature representation of VFs; (2) representing the VFs structures obtained by ESMFold as graph data structures; (3) learning the features of the two modalities through sequence and structure channels respectively; (4) predicting the category of VFs through an MLP.

### Extracting initial protein sequence features

The ESM-2 model is primarily built on the Transformer architecture. The version we use contains 650 million parameters and stacks 33 layers of Transformer encoders during training. Scholars propose that such large-scale pre-trained models, due to their vast number of parameters, exhibit emergent behavior when processing massive amino acid sequences. This behavior enables the model to capture potential structure, functional, and evolutionary information within amino acid sequences [33].

For each input protein sequence, the ESM-2 model generates a 1280-dimensional feature representation for each amino acid. Therefore, the feature representation for each virulence protein sequence is $X^{(0)} \in R^{L \times 1280}$, where $X^{(0)}$ denotes the embeddings of each sequence generated by ESM-2, R denotes set of real numbers and $L$ denotes the length of the VF sequence. These high-dimensional embeddings not only encapsulate the intrinsic characteristics of the sequences but also embed biological significance automatically extracted through deep learning, thereby providing a robust representational foundation for downstream tasks.

### Modelling the graph data structure of VFs

First, we utilized ESMFold to predict the three-dimensional structure of the VFs. Next, we constructed the contact map by considering pairs of Carbon alpha ($C\alpha$) atoms that are within 15 Å of each other. Later, we used additional atomic information to calculate the geometric features of the protein's nodes and edges. Subsequently, we use additional atomic information to calculate the geometric features of protein nodes and edges, where the atomic features include dihedral angles, bond angles, radial basis function embedding, node direction, totaling 184 dimensions, and the edge features include radial basis function embedding, edge direction, edge orientation, edge position encoding, totaling 450 dimensions. To enhance the learning effect of node representations, we concatenate the 1024-dimensional features from ProtT5 to the node features.

Therefore, we define the node features as $v_i \in 1 \times 1208$ where $i$ represents the i-th $C\alpha$ node. Then we define the edge features between $v_i$ and $v_j$ as $e_{ij} \in 1 \times 450$, where $i$ and $j$ represent the $i$-th and $j$-th $C\alpha$ nodes, respectively. Thus, we obtain a graph $G = (V, E)$. $V = \{v^i\}_{i=0}^n$ is the set of vertices where $n$ is the number of $C\alpha$ node of the protein sequence. $E = \{e^{ij}\}$ is the set of edges. A brief description of the node and edge features is provided in Table 2, and the detailed calculation methods can be found in the S2 File titled "Detailed Derivation of Geometric Graph Features" in the Supplementary information.

### Extracting local and global sequence features

Traditional sequence processing models, such as LSTM [34] and Gated Recurrent Unit (GRU) [35], mitigate the vanishing gradient problem through gating mechanisms, thereby better capturing long-term dependencies. However, these models still face challenges with very long sequences: as sequence length increases, the effectiveness of the gating mechanisms diminishes, leading to reduced information transmission efficiency and ultimately impacting model performance. For example, in Recurrent Neural Network (RNN) [36], hidden states can only retain information from a limited number of time steps, making it difficult to effectively capture long-term dependencies. In contrast, we choose to use CNNs and Transformers for extracting protein sequence features. CNNs expand their receptive fields through multiple convolutional layers

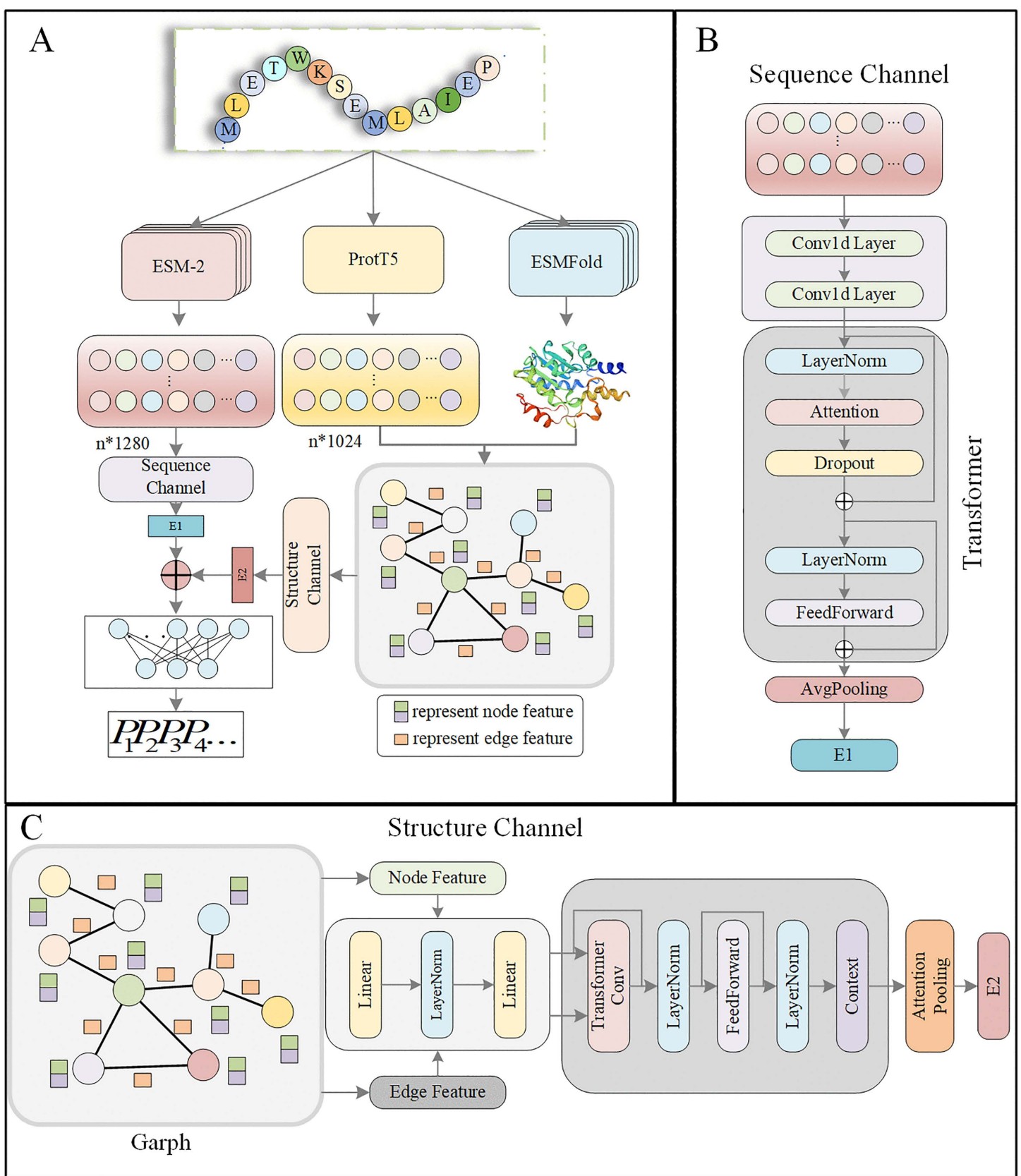

**Fig 3. Framework of PLM-GNN model.** A) The PLM-GNN model adopts a dual-channel architecture to extract sequence and structure features; B) In the sequence channel, a CNN extracts local features, while a Transformer captures global features. The final sequence embedding is obtained through average pooling. C) In the structure channel, node and edge features are first aligned in dimension via linear transformations. They are then refined through a TransformerConv layer and a FeedForward network. Finally, an attention pooling is applied to generate the structure embedding.

**Table 2. The node and edge features for graph representation.**

| | Feature | Description | Dimension |
|---|---|---|---|
| Node features | Dihedral Angles | 'Phi (φ)', 'Psi (ψ)', 'Omega (ω)' | 6 |
| | Bond Angles | 'Alpha (α)', 'Beta (β)', 'Gamma (γ)' | 6 |
| | Radial Basis Function Embedding | 'Cα-N', 'Cα-C', 'Cα-O', 'N-C', 'N-O', 'O-C', 'R-N', 'R-Cα', 'R-C', 'R-O' | 160 |
| | Node direction | Node direction features in the local coordinate system, based on the normalized vectors from Cα to N, C, O, and R. | 12 |
| | ProtT5 | Embeddings generated by the pretrained model | 1024 |
| Edge features | Radial Basis Function Embedding | 'N-N', 'N-Ca', 'N-C', 'N-O', 'N-R', 'Cα-N', 'Cα-Cα', 'Cα-C', 'Cα-O', 'Cα-R', 'C-N', 'C-Cα', 'C-C', 'C-O', 'C-R', 'O-N', 'O-Cα', 'O-C', 'O-O', 'O-R', 'R-N', 'R-Cα', 'R-C', 'R-O', 'R-R' | 400 |
| | Edge Direction | Direction features of the edges, based on normalized vectors from Cα to N, C, O, and R, for both directions (i to j and j to i). | 30 |
| | Edge Orientation | Orientation features of the edges, represented as quaternions. | 4 |
| | Edge Position Encoding | The relative position of edges by calculating the direction difference between edges and combining it with sine and cosine functions of different frequencies. | 16 |

to efficiently capture local features [37,38]; Transformers leverage self-attention mechanisms, allowing each position to directly connect with all other positions without being constrained by sequence length, excelling particularly in capturing long-distance dependencies. Therefore, we will comprehensively extract VF sequence features from both local and global perspectives.

For the original feature sequence $X^{(0)}$ that denotes the embeddings of each sequence generated by ESM-2, we first use a two-layer one-dimensional convolution to extract local features. The definition of extracting local features using one-dimensional convolution is as follows:

$$X^{(1)} = Conv1D(X^{(0)}, W_1), \tag{1}$$

$$X^{(2)} = Conv1D(X^{(1)}, W_2), \tag{2}$$

where $W_1$, $W_2$ represents the learnable parameters (the convolutional kernel) and $Conv1D$ represents Convolutional 1D. Then we apply LayerNorm to the features $X^{(2)}$ obtained from the two-layer CNN, resulting in $X^{(3)}$, which is fed into the multi-head attention mechanism to capture global patterns of the VF sequence. Here, LayerNorm indicates Layer Normalization. The mathematical formulation of multi-head attention is defined as:

$$X^{(3)} = LayerNorm(X^{(2)}), \tag{3}$$

$$Q_i = X^{(3)}W_{Q^i}, K_i = X^{(3)}W_{K^i}, V_i = X^{(3)}W_{V^i}, \tag{4}$$

$$Attention(Q_i, K_i, V_i) = softmax(\frac{Q_i K_i^T}{\sqrt{d_k}})V_i, \tag{5}$$

$$head_i = Attention(Q_i, K_i, V_i), \tag{6}$$

$$X^{(4)} = MultiHead(X^{(3)}) = Concat(head_i, head_2, ..., head_n)W_3 + b_3 \tag{7}$$

where $W_{Q^i}$, $W_{K^i}$, $W_{V^i}$, $W_3$, $b_3$ represents the learnable parameters and $d_k$ represents the dimension of $K_i$. The subsequent computational step integrates nonlinear transformations via activation functions, facilitating the model's ability to capture intricate feature representations. The mathematical formulation of the FeedForward layer is defined as:

$$X^{(5)} = X^{(2)} + Dropout(X^{(4)}), \tag{8}$$

$$X^{(6)} = Dropout(RELU(LayerNorm(X^{(5)})W_4 + b_4)W_5 + b_5), \tag{9}$$

$$E_S = Avgpooling(X^{(5)} + X^{(6)}), \tag{10}$$

where $W_4$, $b_4$, $W_5$, $b_5$ represents the learnable parameters, *Dropout* denotes a technique for randomly deactivating neurons during training to prevent overfitting, *RELU* refers to the rectified linear unit activation function that introduces non-linearity, and *Avgpooling* signifies the average pooling operation. And $E_S$ denotes the final output embedding from the sequence channel.

## Extracting structure features

Geometric graph neural networks demonstrate high efficacy in modeling the inherent structural characteristics of proteins, such as bond lengths, angles, and dihedral angles [39,40], which are essential for understanding the spatial relationships within a protein's 3D structure [41]. By incorporating such geometric features, geometric graph neural networks provide a more detailed representation of a protein's topological structure and its interactions with other molecules [42]. The integration of geometric graph neural networks with ESMFold-derived structures facilitates a thorough exploration of protein topology, thereby improving our ability to predict and analyze protein function with greater precision and depth [43]. Remarkably, geometric graph neural networks have exhibited outstanding efficacy in diverse computational tasks [44–46].

Therefore, we employed geometric graph neural networks to extract structure features. For the node and edge features obtained from the protein's 3D structure and the node features derived from ProtT5, which together form the $G = (V, E)$, we first input the edge and node features into an encoder module to obtain the node and edge embeddings. The formula for this process is as follows:

$$h_v^i = LayerNorm(W_6 v_i + b_6)W_7 + b_7, \tag{11}$$

$$h_e^{ij} = LayerNorm(W_8 e_{ij} + b_8)W_9 + b_9, \tag{12}$$

where $W_6$, $b_6$, $W_7$, $b_7$, $W_8$, $b_8$, $W_9$, $b_9$ represents the learnable parameters. $h_v^i$ and $h_e^{ij}$ represent the encoded features, both with a dimensionality of 256. Then we use Graph Transformer [47] to extract structure information from the obtained graph, with the specific formula as follows:

$$x_i = W_{10} h_v^i + \sum_{j \in N(i)} \alpha_{i,j}(W_{11} h_v^j + W_{12} e_{ij}), \tag{13}$$

$$\alpha_{i,j} = softmax(\frac{(W_{13}h_v^j)^T(W_{14}h_v^j + W_{12}e_{ij})}{\sqrt{d}}),$$
(14)

where $W_{10}$, $W_{11}$, $W_{12}$, $W_{13}$, $W_{14}$ represents the learnable parameters. $\alpha_{i,j}$ represents the attention coefficients. $N(i)$ represents the set of all nodes that are connected to $node_i$ by an edge. The extracted features are propagated through residual connections and a FeedForward neural network, followed by feature refinement via the context module. The corresponding mathematical expressions are defined below:

$$x_i^1 = LayerNorm(h_v^i + Dropout(x_i)),$$
(15)

$$FeedForword : x_i^2 = (RELU(W_{15}x_i^1 + b_{15}))W_{16},$$
(16)

$$x_i^3 = LayerNorm(x_i^1 + Dropout(x_i^2)),$$
(17)

$$context : \begin{cases} E_i = AvgPooling(x_i^3) \\ x_i^4 = x_i^3 \times Sigmoid(RELU(W_{17}E_i + b_{17})W_{18} + b_{18}), \end{cases}$$
(18)

where $W_{15}$, $b_{15}$, $W_{16}$, $b_{16}$, $W_{17}$, $b_{17}$, $W_{18}$, $b_{18}$ represents the learnable parameters and *sigmoid* denotes the activation function that maps outputs to [0,1]. Finally, we obtain the structure embedding through attention pooling, as given by the following formula:

$$E_i' = AttentionPooling(x_i^4),$$
(19)

### Classifying VFs and weighted cross-entropy loss

We compared three fusion strategies for integrating the sequence features and structure features learned through the dual-channel feature extraction module. The first strategy was concatenation, which combines features from different sources along the feature dimension to form a longer fused feature vector. The second strategy was direct addition, where the features obtained from the two channels are directly summed for fusion. The third strategy employed an attention mechanism, which dynamically computes the weights of sequence features and geometric structural features, achieving adaptive feature combination through weighted fusion. As shown in S5 Fig in S3 File, the direct addition strategy yielded the best performance, and was therefore ultimately adopted for multi-channel feature fusion. Specifically, the embeddings derived from the two parallel channels are summed, followed by a MLP for final classification of the VFs. The corresponding mathematical formulation is provided below:

$$prob = softmax(MLP(E_i' + E_S)),$$
(20)

To optimize the classification performance, we carefully design the loss function to address potential challenges in the training process. In conventional multi-class classification tasks, cross-entropy loss is widely adopted as the standard loss function. However, our dataset exhibits substantial variation in sample distribution across categories, potentially causing model predictions to favor majority classes. To mitigate this imbalance, we implemented weighted cross-entropy loss [48], which assigns category-specific weights to enhance the model's focus on underrepresented samples. The mathematical formulation of the loss function is provided below:

$$L_{wce} = -\frac{1}{N} \sum_{n=1}^{N} \sum_{c=0}^{C-1} w'_c \times I(y_i = c) \times \log(p_i(c)), \tag{21}$$

where $N$ represents the total number of samples in the dataset, $C$ denotes the total number of classes, and $w_c$ is the weight assigned to class $c$, which helps adjust the importance of each class based on its frequency. The actual class label of the $i$-th sample is denoted by $y_i$, whereas $p_i(c)$ represents the model's estimated probability that the $i$-th sample is classified under class $c$. Lastly, $I(y_i = c)$ is an indicator function that takes the value 1 when the true label $y_i$ matches the class $c$, and 0 otherwise. For each class $c$, the weight $w'_c$ is calculated as follows:

$$w_c = \frac{N}{count_c + \varepsilon}, \tag{22}$$

$$w'_c = \frac{w_c}{\sum_{j=1}^{C} w_c}, \tag{23}$$

where $count_c$ is the number of occurrences of class $c$, $\varepsilon$ is a very small value (1e−6), used to prevent division by zero.

## Results

### Evaluation metrics

During the validation and independent testing stages, we employed a diverse set of evaluation metrics to comprehensively analyze model performance, comprising Accuracy (ACC), confusion matrix analysis, and Receiver Operating Characteristic (ROC) curve evaluation. To address dataset imbalance, weighted variants of Accuracy, Recall, and F1-score were adopted to ensure balanced assessment of model efficacy. The mathematical definitions for ACC, Precision, Recall, and F1-score used in this study are provided below:

$$ACC = \frac{TP + TN}{TP + TN + FP + FN}, \tag{24}$$

$$Precision_i = \frac{TP}{TP + FP}, Precision_w = \frac{\sum_{i}^{C}(Support_i \times Precision_i)}{\sum_{i}^{C} Support_i}, \tag{25}$$

$$Recall_i = \frac{TP}{TP + FN}, Recall_w = \frac{\sum_{i}^{C}(Support_i \times Recall_i)}{\sum_{i}^{C} Support_i}, \tag{26}$$

$$F1_i = \frac{2 \times Precision_i \times Recall_i}{Precision_i + Recall_i}, F1_w = \frac{\sum_{i}^{C}(Support_i \times F1_i)}{\sum_{i}^{C} Support_i}, \tag{27}$$

where $i$ denote the $i$-th class, $C$ is the number of classes, $Support_i$ is the number of samples in class $i$, and $TP$, $TN$, $FP$, and $FN$ represent the counts of true positives, true negatives, false positives, and false negatives, respectively. $i$ denote the $i$-th class. All metrics are calculated using the sklearn [49] package.

### Parameter settings

The PLM-GNN model was trained and tested on an independent GeForce RTX 4090 24 GB GPU. During training, the best parameters were selected based on the minimum validation loss. Furthermore, the hyperparameters of our model were configured as

follows: a learning rate of 0.00005, a batch size of 32, and a fixed random seed of 42. In terms of model architecture, we utilized a single Transformer layer with a 4-head multi-head attention mechanism, a dropout rate of 0.3, and a single-layer MLP. During training, we adopted the Cosine Annealing strategy as the learning rate scheduler [50] to dynamically adjust the learning rate. This approach facilitates faster convergence in the initial training phase while maintaining stable optimization in later stages, thereby reducing the risk of the model converging to suboptimal local minima and improving its generalization performance. To further enhance model generalization, an early stopping criterion with a patience parameter of 5 was integrated into the training protocol to mitigate overfitting risks.

## The performance of PLM-GNN on independent test sets

To comprehensively assess the VF classification capability of our model, a dual evaluation framework incorporating both validation and independent test sets was utilized. During training, the most effective parameter configurations were iteratively retained based on validation set performance. This approach guaranteed robust generalizability, enabling the model to attain near-optimal predictive accuracy on previously unencountered data instances.

The experimental results of our proposed multi-classification method for VFs, termed PLM-GNN, are shown in Fig 4A. The results demonstrate that the model achieved an accuracy of 88.56% on the validation set and also performed well on the independent test set, with an accuracy of 86.47%. To provide a more intuitive reflection of the model's overall performance, we plotted a confusion matrix as Fig 4B, which displays the number of correctly and incorrectly predicted samples for each category. From the confusion matrix, it can be observed that the model correctly predicts a large number of samples for the top six VF categories. However, for the *Biofilm* category, due to its limited sample size, the model's predictive performance for this category is relatively lower, highlighting the impact of data imbalance on model performance.

To systematically validate the model's discriminative capacity across diverse VF categories, ROC curves were generated (Fig 4C). All categories achieved AUC values exceeding 0.9, demonstrating robust inter-class differentiation capability in multi-class scenarios. This quantitative evidence substantiates the model's superior classification performance and operational reliability. Notably, despite the small sample size of the *Biofilm* category, its corresponding ROC curve still exhibited a high degree of discriminative capability.

## Comparison with other methods in terms of VF classification

To comprehensively evaluate the performance of PLM-GNN, we selected several sequence processing models and graph neural network models for comparison. The sequence processing models included CNN [31], GRU, LSTM [35],

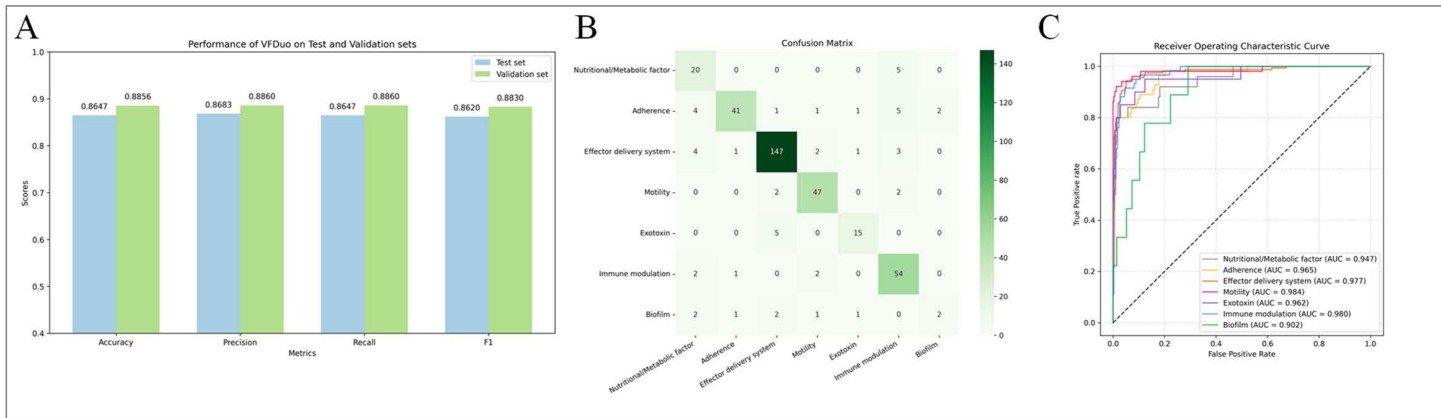

**Fig 4. The performance of PLM-GNN from three perspectives.** (A) Comparative evaluation of PLM-GNN's classification metrics across validation and test datasets. (B) The confusion matrix of PLM-GNN on the test set. (C) The ROC curves for each type of VF by PLM-GNN on the test set.

Bidirectional Long Short-Term Memory (BiLSTM) [51], Transformer [52,53], CNN-LSTM, CNN-BiLSTM, and CNN-GRU. The graph neural network models included GCN [27], Graph Attention Network (GAT) [54], and Graph Transformer [48]. For the sequence processing models, we utilized features generated by ESM-2. These features were input into the sequence models to extract higher-level representations, which were then passed through an MLP for classification. For the graph neural network models, we employed features generated by ProtT5, which were concatenated with geometric features derived from PDB to form a combined feature set. These features were processed by the graph neural network models to extract relevant information, followed by classification using an MLP. Subsequently, we performed hyperparameter tuning on these baseline models. The detailed tuning ranges and the final selected parameters are summarized in S1 Table in S3 File, while the corresponding performance on the test set is visualized in Table 3. To comprehensively evaluate the performance of the proposed model, we selected the two best-performing baseline models from the test set and conducted a statistical significance analysis for PLM-GNN. The independent test set was partitioned into five equal subsets, with four subsets used for testing in each iteration, producing five distinct evaluation datasets. Independent samples t-tests were performed on these datasets. As illustrated in Fig 5, PLM-GNN exhibited consistent and statistically significant advantages over both GCN and GAT across multiple performance metrics (all p-values < 0.05), confirming its statistical superiority over these baseline models.

To further evaluate the models' ability to distinguish between classes, we analyze the clustering patterns in the t-SNE visualizations. As depicted in Fig 6, models such as GAT, CNN-LSTM, CNN-BiLSTM, CNN-GRU, LSTM and BiLSTM exhibit poorly defined class boundaries, suggesting limited capability in accurately differentiating classes, which consequently reduces classification accuracy. Similarly, the t-SNE visualizations of Graph Transformer, GCN, GRU, CNN, LSTM and Transformer reveal overlapping clusters of samples from multiple classes, indicating challenges in correctly assigning samples to their respective classes. In contrast, the t-SNE visualization of PLM-GNN shows well-separated class clusters, underscoring its enhanced ability to effectively integrate features from both sequential and structure channels. This result highlights the clear advantage of PLM-GNN in classification tasks.

## Selection of protein pretrained language models

Among the currently popular protein pretrained language models are ESM-2, ProtT5, and ProtBert [20]. In this study, we conducted a feature selection experiment based on a VF multi-class classification model to determine the most effective representation combination for this task. Specifically, each of the three pretrained models was independently applied to the model's two input channels: the sequence channel and the structural channel. The evaluation results are presented in

Table 3. Comparison results of different methods on the independent test.

| Method | Acc | Pre | Recall | F1 |
| --- | --- | --- | --- | --- |
| GCN | 0.840 | 0.844 | 0.841 | 0.832 |
| GAT | 0.840 | 0.848 | 0.841 | 0.841 |
| Graph Transformer | 0.827 | 0.829 | 0.828 | 0.821 |
| CNN | 0.828 | 0.830 | 0.828 | 0.823 |
| GRU | 0.833 | 0.837 | 0.833 | 0.830 |
| LSTM | 0.828 | 0.829 | 0.828 | 0.825 |
| BiLSTM | 0.835 | 0.841 | 0.836 | 0.830 |
| Transformer | 0.833 | 0.841 | 0.833 | 0.832 |
| CNN-LSTM | 0.832 | 0.838 | 0.833 | 0.830 |
| CNN-BiLSTM | 0.812 | 0.817 | 0.812 | 0.809 |
| CNN-GRU | 0.832 | 0.833 | 0.833 | 0.830 |
| PLM-GNN | **0.864** | **0.868** | **0.864** | **0.862** |

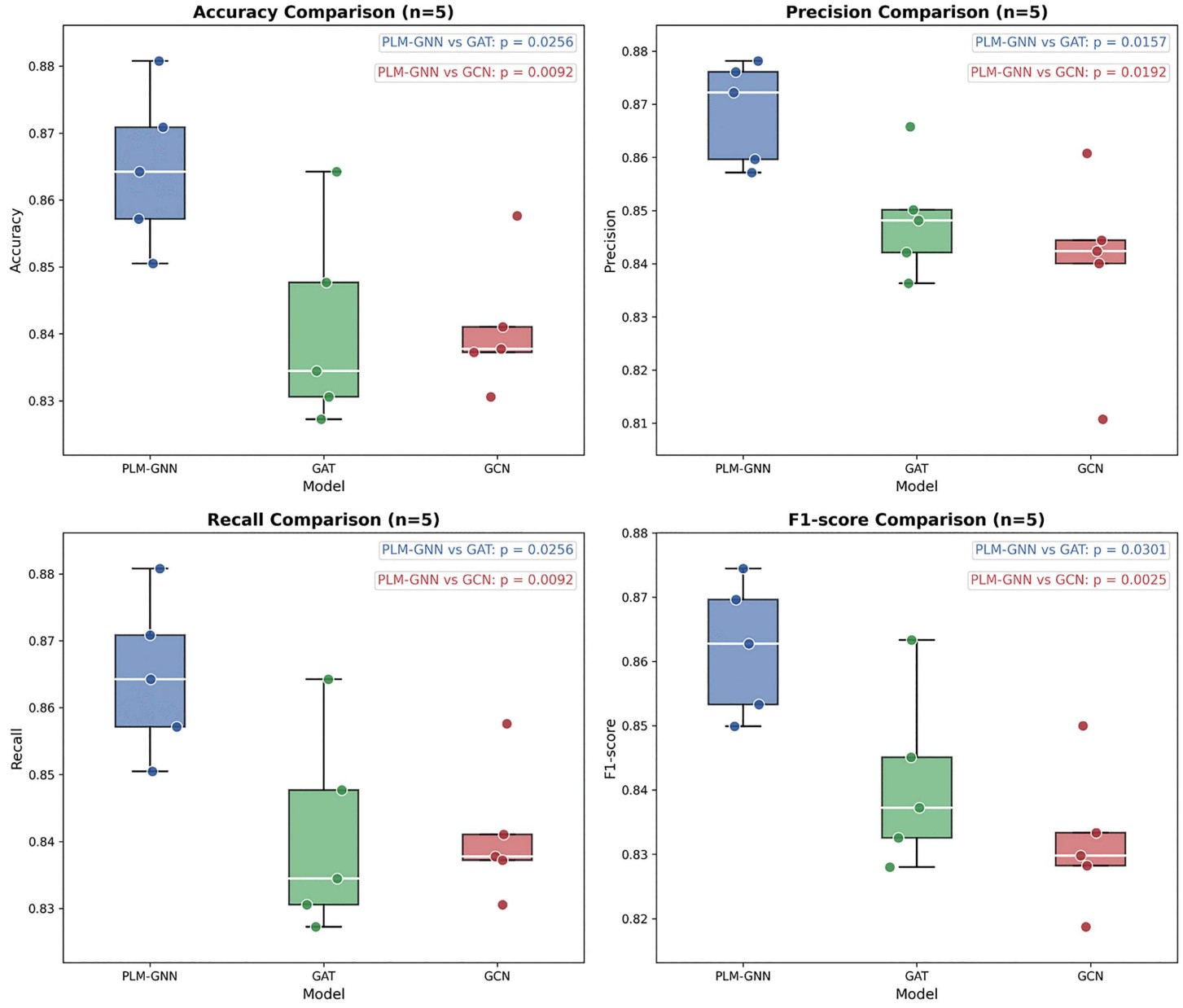

**Fig 5. Statistical significance comparison of PLM-GNN with GCN and GAT in VF classification.**

Table 4, where the first part of each combination refers to the model used for the sequence channel, and the second part indicates the node features used in the structural channel. These results on the independent test set demonstrate that employing ESM-2 for the sequence channel and ProtT5 for the structural channel achieves the best performance. Accordingly, this combination of pretrained language models was selected for the final model configuration.

## Ablation experiments

To validate the effectiveness of the dual-channel architecture in our model, we perform ablation experiments to assess two model variants: (1) PLM-GNN -w/o Transformer: which substitutes the Transformer module with an MLP in the sequence

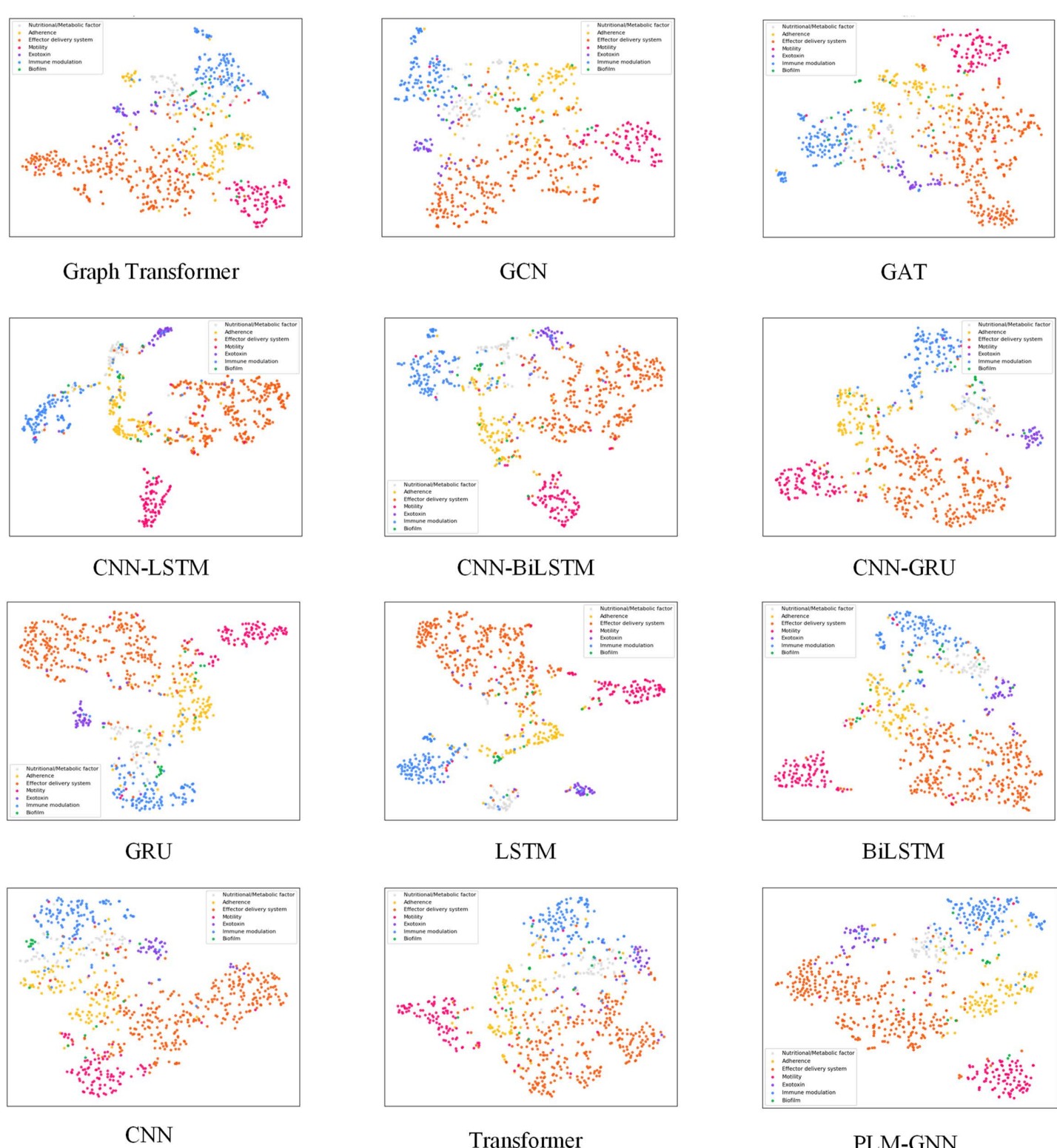

**Fig 6. The t-SNE clustering results of embeddings generated by PLM-GNN and other methods.**

**Table 4. Combinations of pretrained language models and the performance.**

| PLM Combination | Acc | Pre | Recall | F1 |
|---|---|---|---|---|
| ProtBert-ESM2 | 0.822 | 0.824 | 0.822 | 0.819 |
| ProtBert-ProtBert | 0.788 | 0.780 | 0.788 | 0.782 |
| ProtBert- ProtT5 | 0.822 | 0.830 | 0.822 | 0.818 |
| ProtT5 -ESM2 | 0.846 | 0.855 | 0.846 | 0.845 |
| ProtT5 - ProtT5 | 0.851 | 0.860 | 0.851 | 0.850 |
| ProtT5 -ProtBert | 0.809 | 0.808 | 0.809 | 0.806 |
| ESM2-ESM2 | 0.840 | 0.848 | 0.841 | 0.841 |
| ESM2-ProtBert | 0.833 | 0.833 | 0.833 | 0.829 |
| ESM2- ProtT5 | **0.864** | **0.868** | **0.864** | **0.862** |

channel while maintaining the original architecture of the structure channel; (2) PLM-GNN -w/o CNN: which replaces the CNN module with an MLP in the sequence channel, preserving the structure channel; (3) PLM-GNN -w/o Seq: which removes the entire sequence channel and exclusively utilizes structure channel and (4) PLM-GNN -w/o GNN: eliminates the structure channel and retains only the sequence channel for feature learning. The experimental results presented in Table 5 demonstrate that the full PLM-GNN model consistently outperforms these ablated variants. Notably, removing the Transformer module leads to a marked drop in performance, highlighting its critical role in capturing the global contextual features of VFs. Similarly, the exclusion of the CNN module results in a significant performance decline, underscoring its importance in extracting local sequence-level patterns. Furthermore, the full model surpasses the PLM-GNN-w/o Seq variant, emphasizing the essential contribution of protein sequence information to the identification of complex internal patterns and key discriminative features. In addition, the full model also outperforms the PLM-GNN-w/o GNN variant, which highlights the significance of structural information in modeling the three-dimensional conformation of proteins and their internal interactions. These findings collectively confirm that each module plays an indispensable role in achieving optimal performance.

## Interpretability of the learned representations via t-SNE visualization

To comprehensively investigate the learning mechanisms and feature extraction capabilities of the PLM-GNN model, we first introduced its dual-channel architecture. The PLM-GNN model processes features from two distinct sources through independent channels: one channel focuses on sequence features extracted by ESM-2, while the other handles structure features generated by a geometric graph neural network. This design enables the model to capture critical information from diverse perspectives, thereby facilitating a more holistic understanding of VF characteristics.

Initially, we employed t-SNE [55] a widely used dimensionality reduction technique, to project the sequence features obtained from ESM-2 into a two-dimensional space for visualization. As illustrated in Fig 7A, the original features in this

**Table 5. Ablation experiment results on the PLM-GNN.**

| Method | Test set | | | | Validation set | | | |
|---|---|---|---|---|---|---|---|---|
| | Acc | Pre | Recall | F1 | Acc | Pre | Recall | F1 |
| PLM-GNN -w/o Transformer | 0.7639 | 0.7890 | 0.7640 | 0.7650 | 0.7766 | 0.7980 | 0.7770 | 0.7760 |
| PLM-GNN -w/o CNN | 0.8276 | 0.8260 | 0.8280 | 0.8220 | 0.8484 | 0.8510 | 0.8480 | 0.8480 |
| PLM-GNN -w/o Seq | 0.7374 | 0.7630 | 0.7370 | 0.7280 | 0.7926 | 0.8020 | 0.7930 | 0.7900 |
| PLM-GNN -w/o GNN | 0.8329 | 0.8320 | 0.8330 | 0.8300 | 0.8617 | 0.8610 | 0.8620 | 0.8600 |
| PLM-GNN | **0.8647** | **0.8683** | **0.8647** | **0.8620** | **0.8856** | **0.8860** | **0.8860** | **0.8830** |

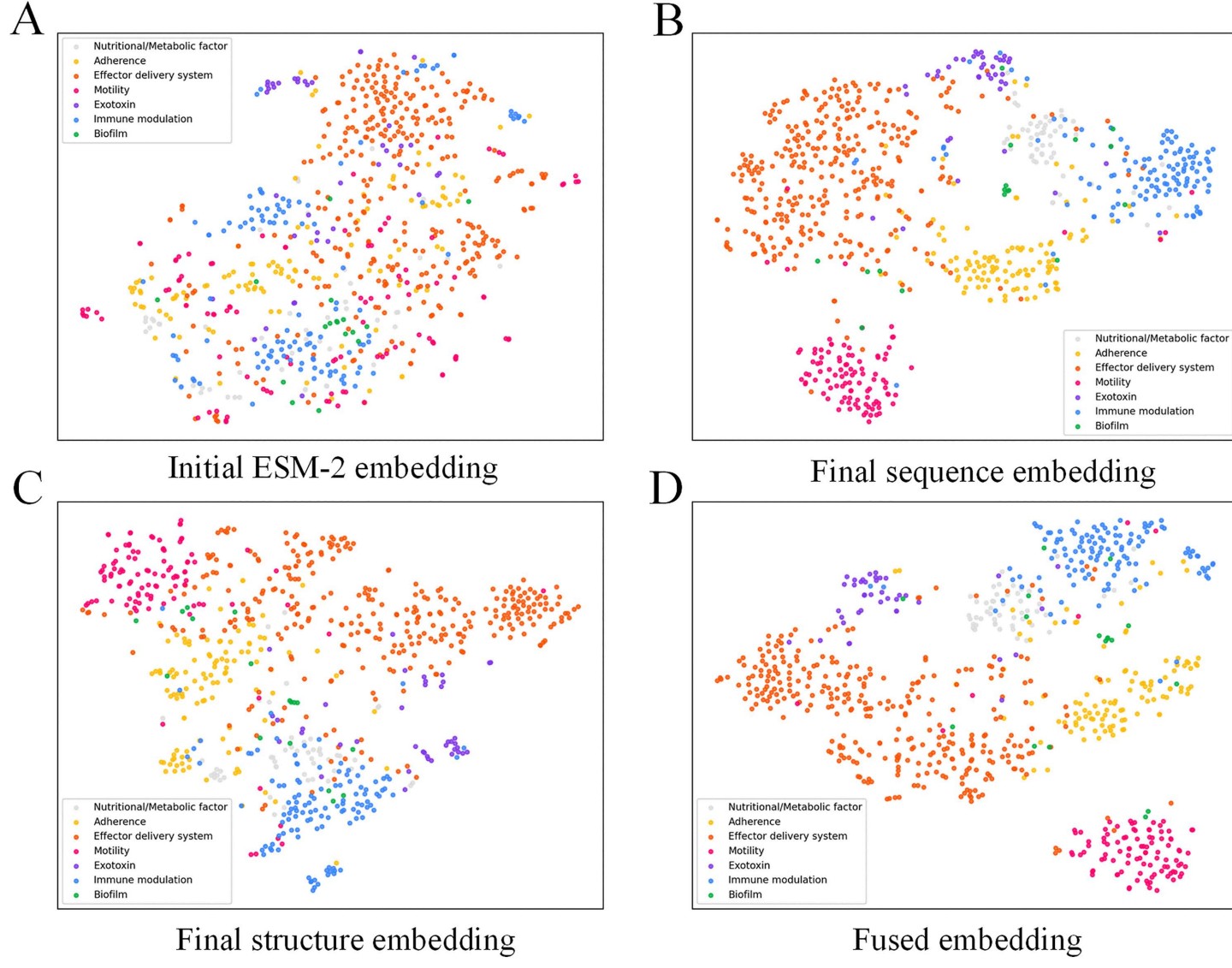

**Fig 7. The t-SNE visualization of different embeddings.**

reduced-dimensional space exhibit overlapping distributions, with sample points from different categories intermixed and indistinguishable. This suggests that the original features alone are inadequate for clearly differentiating between various VF categories. In contrast, when the learned representations from the trained PLM-GNN model were projected into a two-dimensional space using t-SNE (Fig 7D), the separation between categories improved significantly, with distinct clustering patterns emerging. This outcome demonstrates that the PLM-GNN model effectively learns discriminative features suitable for VF prediction, thereby enhancing classification performance.

Further analysis of the individual performances of the two channels (Fig 7B and 7C) reveals that each channel independently achieves clear classification of VFs after processing its respective features. This not only highlights the efficacy of the two channels but also underscores their complementary roles in feature extraction. Collectively, these experiments demonstrate the effectiveness of the PLM-GNN model in extracting meaningful information from complex raw features

and learning advanced representations that improve classification performance, while also validating the dual-channel design.

To further validate the interpretability of the PLM-GNN model, we conducted a statistical experiment based on attention scores extracted from the sequence channel. We selected the T4SS within the dataset for analysis. Initially, we filtered out the top 5 amino acids with the highest attention scores in each sequence. Subsequently, we conducted a statistical analysis of the overall distribution of these high-frequency amino acids across the entire T4SS category. The results showed that the five most frequently occurring amino acids were phenylalanine (F), threonine (T), proline (P), serine (S), and glycine (G). These amino acids play critical roles in the biological functions of the T4SS system. Among them, F is widely present in the hydrophobic motifs at the C-terminus of effector proteins, mediating binding with the LvgA protein and subsequent transport processes [56]; T and S serve as important phosphorylation sites involved in regulating T4SS assembly; while P and G, as structure-stabilizing amino acids, play key roles in the transmembrane channel structure of T4SS [57]. These findings consistently demonstrate that the attention mechanism of the PLM-GNN model can effectively capture key amino acid features related to transport, regulation, and structure in the T4SS system, reflecting its strong biological interpretability.

## Analysis of potential causes for model prediction errors

To investigate the causes of model prediction errors, in the S1 File we computed both Pearson and Spearman correlation coefficients between the misprediction label vector of the PLM-GNN and the Predicted Local Distance Difference Test (pLDDT) [25] score vector. The results indicated no significant correlation between PLM-GNN classification errors and pLDDT scores.

Next, we examined the influence of species on prediction errors. As shown in S2 Fig in S3 File, no particular species exhibited notably prominent misprediction. We then further analyzed whether the length of virulence factor sequences is associated with prediction errors. Sequences were grouped at intervals of 200 in length, and the performance metrics for each group were calculated, as shown in S3 Fig in S3 File. The results revealed that the performance of sequences between 1000 and 1240 in length was not significantly lower than that of sequences in the 200–400 range, indicating that prediction errors are not related to sequence length.

Additionally, through t-SNE visualization of the fused features from the model (Fig 7D), we observed that samples from some different categories clustered together in the embedding space, indicating feature similarity. For instance, the features of the *Immune modulation* and *Nutritional/Metabolic factor* categories were closely distributed. It can therefore be inferred that the model's prediction errors primarily stem from the proximity of certain categories in the feature space and their representational similarity, resulting in insufficient inter-category distinguishability.

## Applied to effector delivery system classification

*Effector delivery system* is specialized mechanisms used by bacteria to transport proteins, known as effectors, into host cells or the extracellular environment. These systems play crucial roles in bacterial pathogenesis and symbiosis by facilitating interactions between bacteria and their hosts [58,59]. The recognition of *Effector delivery system* is crucial as it uncovers how pathogens manipulate host cells to facilitate infection, thereby providing key insights for developing new therapeutic strategies, vaccines, and biological defense measures. Moreover, by predicting the categories of *Effector delivery system*, we can also validate the effectiveness of our model in classifying VFs.

We first extracted the sequences belonging to the *Effector delivery system* category from the multi-classification dataset, resulting in a total of 1,580 sequences. Due to the limited number of sequences in the T5SS category, we focused our classification analysis on the five categories of T2SS, T3SS, T4SS, T6SS, and T7SS. After filtering, the sequence counts for T2SS, T3SS, T4SS, T6SS, and T7SS were 86, 649, 561, 206, and 71, respectively, amounting to a total of 1,573 sequences. These sequences were then divided into training, test, and validation sets in an 8:1:1 ratio. The detailed distribution of the data is presented in Table 6.

From the confusion matrix in Fig 8A, it is evident that the model performs well in predicting the categories of T2SS, T3SS, T4SS, and T6SS. However, its performance for the T7SS category is slightly weaker, primarily due to the limited number of samples in this category. Furthermore, the ROC curve in Fig 8B demonstrates that the model can accurately distinguish among the various categories in the *Effector delivery system* dataset. By leveraging the complementary strengths of sequence and structure data, PLM-GNN not only significantly enhances the model's discriminative capabilities but also ensures its broad applicability across diverse datasets.

## Performance on binary classification of VFs

Since the research on VFs usually requires first determining whether a protein is a VF and then conducting more detailed classification, we retrained and evaluated the model on a binary classification dataset to ensure that a complete VF classification process could be achieved through our model. Here, we used the DeepVF [15] dataset, from which we selected and obtained a dataset containing 2,873 VFs and 2,872 non-VFs. Subsequently, we divided it into a training set and a validation set in an 8:2 ratio. Additionally, to better compare performance with other models, we chose the DeepVF test set for testing.

**Table 6. The distribution of the *Effector delivery system* dataset.**

| classes | Train set | Valid set | Test set | Total |
|---------|-----------|-----------|----------|-------|
| T2SS | 68 | 9 | 9 | 86 |
| T3SS | 519 | 65 | 65 | 649 |
| T4SS | 448 | 56 | 57 | 561 |
| T6SS | 164 | 21 | 21 | 206 |
| T7SS | 56 | 7 | 8 | 71 |

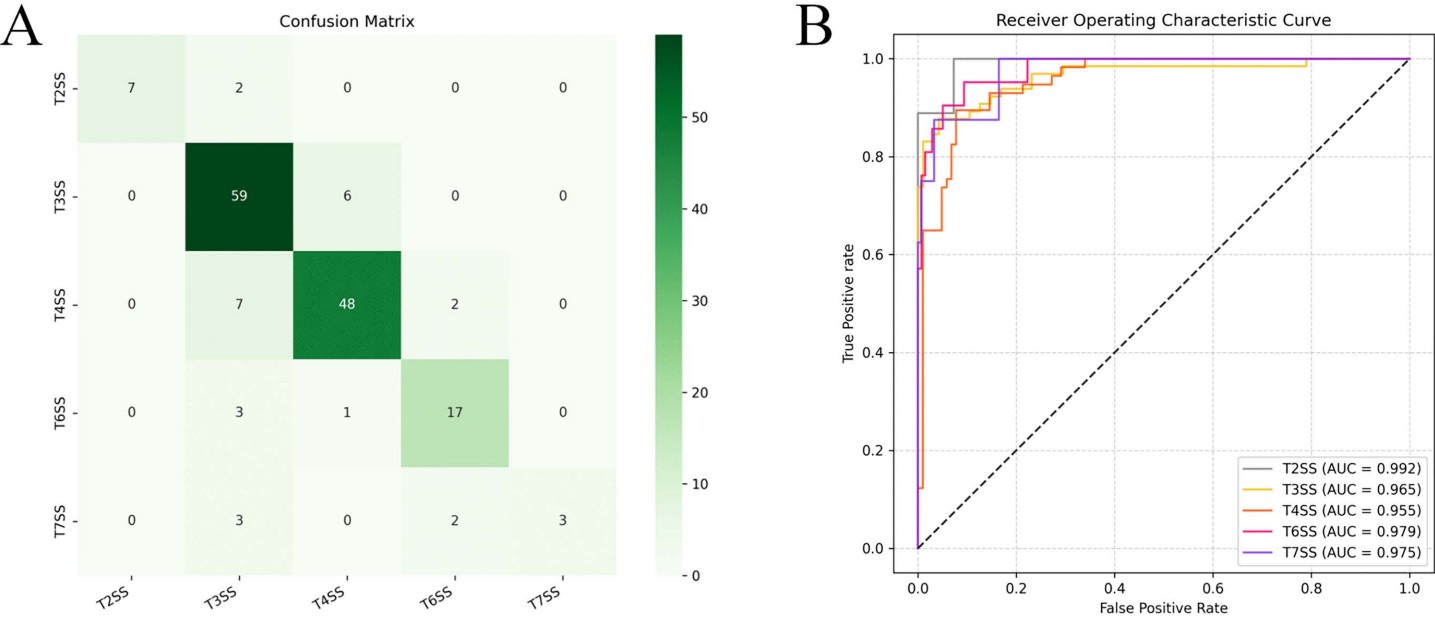

**Fig 8. The performance on the test set for the five types of *Effector delivery system*.**

We then compared PLM-GNN with several state-of-the-art VF identification models on the same test set, including BLAST [11], MP3 [13], PBVF [14], VirulentPred [12], DeepVF, VF-Pred [16], DTVF [23] and GTAE-VF [26]. As shown in Table 7, PLM-GNN achieved higher accuracy scores than the other eight models, with accuracy improvements of 11.2%, 20.2%, 6.8%, 25.5%, 5.0%, 2.7%, 1.6%, and 1.3%, respectively. Overall, our model demonstrates consistently superior performance across key evaluation metrics, underscoring the effectiveness of the proposed dual-channel architecture.

## Accurately classifying VFs with remote homology

In biochemistry, the function of a protein is predominantly governed by its three-dimensional structure rather than its amino acid sequence alone. Certain proteins exhibit low sequence similarity while sharing structure similarities, resulting in analogous functional characteristics. Structure alignment of proteins offers profound insights into functional relationships across extended evolutionary distances, a feat often unachievable through sequence-based alignment methods [60,61]. Protein pairs that have dissimilar sequences but comparable structures—identified by a sequence identity of less than 0.3 and a TM-score exceeding 0.5—are referred to as "remote homologs." [62–64].

As demonstrated in Fig 9, the PLM-GNN model exhibits a strong capability to learn from structurally similar VFs and identify remote homology even under low sequence similarity, enabling accurate functional classification. This highlights

**Table 7. Comparison of PLM-GNN's performance with existing methods on the VF identification task using the same test set.**

| Methods | Accuracy | F1-score | Specificity | Sensitivity |
|---|---|---|---|---|
| BLAST | 0.750 | 0.732 | 0.818 | 0.682 |
| MP3 | 0.660 | 0.612 | 0.783 | 0.536 |
| PBVF | 0.794 | 0.790 | 0.814 | 0.774 |
| VirulentPred | 0.607 | 0.620 | 0.573 | 0.641 |
| DeepVF | 0.812 | 0.807 | 0.833 | 0.790 |
| VF-Pred | 0.835 | 0.825 | 0.760 | **0.870** |
| DTVF | 0.845 | 0.839 | **0.889** | 0.802 |
| GTAE-VF | 0.849 | 0.834 | 0.879 | 0.814 |
| PLM-GNN | **0.862** | **0.857** | 0.886 | 0.838 |

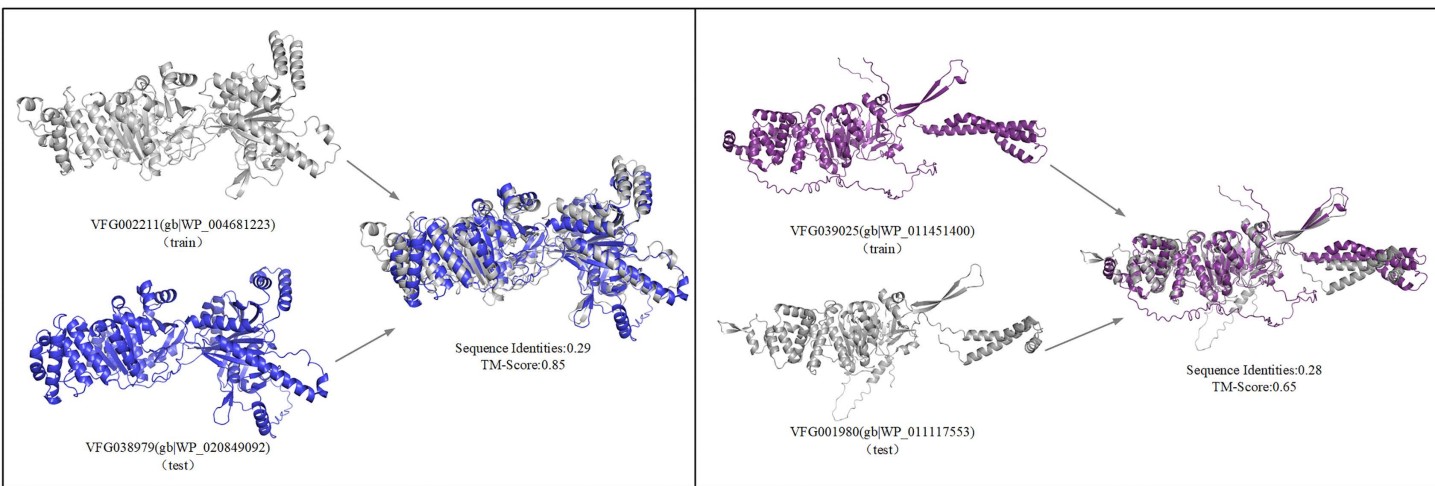

**Fig 9. Identify remote homology VFs.** (Train refers to the *Effector delivery system* type VFs in the training set, while test refers to the *Effector delivery system* type VFs correctly predicted in the test set.).

the model's capacity to extract essential information from 3D protein structures, effectively recognizing and categorizing functionally analogous VFs beyond sequence-based constraints. To quantitatively validate this ability, we constructed a dedicated dataset containing 93 VFs from the test set that show remote homology to sequences in the training set. Our model achieved 91.4% accuracy, correctly predicting 85 out of 93 instances. In comparison, the standalone sequence channel achieved 79 correct predictions, and the standalone structure channel achieved 70. These results demonstrate that the dual-channel architecture—which integrates complementary sequence and structure features—significantly enhances the model's performance in detecting remote homology.

## Discussion and conclusion

This study proposes a novel dual-channel model, PLM-GNN, which independently captures the sequence features and structural features of VFs through a dedicated CNN-Transformer channel and a geometric graph neural network channel, respectively. By integrating the representations from both channels, the model achieved state-of-the-art performance in both multi-class and binary VF classification tasks, surpassing existing methods, as validated through comparative evaluation and t-SNE visualization. PLM-GNN also demonstrated strong performance in predicting effector delivery systems, highlighting its broad application potential in improving protein functional annotation. While the model may facilitate future studies aimed at identifying virulence-related mechanisms or therapeutic targets [65], its immediate contribution is providing a more accurate and comprehensive classification tool for VFs.

However, several important limitations of this study should be noted. First, constrained by the dataset scale, the model was trained only on the seven most abundant VF categories from the VFDB core dataset, excluding seven rarer VF categories. This inevitably limits the model's generalization capability for under-represented VF categories, and its performance on these categories remains unclear. Second, the model underperformed in predicting minority classes such as effector delivery systems (e.g., T7SS), highlighting the challenges posed by highly imbalanced data. Future work will explore solutions such as few-shot learning or employing generative adversarial networks (GANs) to generate samples for rare categories. Third, due to GPU memory constraints, proteins longer than 1240 amino acids were excluded, which may prevent the model from learning long-sequence VFs with complex domains. Future plans include utilizing GPUs with larger memory capacity and exploring strategies such as segmenting long-sequence VFs for processing. Looking ahead, we will focus on expanding the model's application to all categories in VFDB, enhancing the scalability of the model architecture, and integrating features from emerging large language models for amino acid sequences (e.g., Nucleotide Transformer [66]) to further improve performance.

## Supporting information

**S1 File. Distribution of pLDDT and correlation with model performance.**
(DOCX)

**S2 File. Detailed derivation of geometric graph features.**
(DOCX)

**S3 File. Supplemental figures and tables.**
(DOCX)

## Author contributions

**Conceptualization:** guanghui Li, Peiyang Song, Cheng Liang.

**Data curation:** Peiyang Song.

**Funding acquisition:** guanghui Li, Cheng Liang.

**Investigation:** guanghui Li.

**Methodology:** guanghui Li, Peiyang Song.

**Project administration:** guanghui Li, Cheng Liang.

**Software:** Peiyang Song.

**Supervision:** Cheng Liang.

**Validation:** Peiyang Song.

**Visualization:** Peiyang Song.

**Writing – original draft:** guanghui Li, Peiyang Song.

**Writing – review & editing:** Jiawei Luo, Cheng Liang.

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
