## [Decision Letter · Decision Letter 0]

28 Jul 2025

Dear Dr. Li,

Thank you for submitting your manuscript to PLOS ONE. After careful consideration, we feel that it has merit but does not fully meet PLOS ONE’s publication criteria as it currently stands. Therefore, we invite you to submit a revised version of the manuscript that addresses the points raised during the review process.

We look forward to receiving your revised manuscript.

Kind regards,

Aiqing Fang

Academic Editor

PLOS ONE

Journal Requirements:

“This work was supported by the National Natural Science Foundation of China [grant numbers 62362034, 62372279]; the Natural Science Foundation of Jiangxi Province of China [grant number 20232ACB202010]; the Natural Science Foundation of Shandong Province [grant number ZR2023MF119]; and the Jiangxi Province Key Laboratory of Advanced Network Computing [grant number 2024SSY03071].”

Reviewers' comments:

Reviewer's Responses to Questions

**Comments to the Author**

1. Is the manuscript technically sound, and do the data support the conclusions?

Reviewer #1: Yes

Reviewer #2: Partly

2. Has the statistical analysis been performed appropriately and rigorously?

Reviewer #1: Yes

Reviewer #2: Yes

3. Have the authors made all data underlying the findings in their manuscript fully available?

Reviewer #1: Yes

Reviewer #2: No

4. Is the manuscript presented in an intelligible fashion and written in standard English?

Reviewer #1: Yes

Reviewer #2: Yes

Reviewer #1: This manuscript presents PLM-GNN, a novel dual-channel deep learning model for classifying bacterial virulence factors (VFs). The model uniquely integrates a sequence channel, utilizing pre-trained language models and Transformer architectures, with a structure channel that employs a geometric graph neural network on ESMFold-predicted 3D structures. The authors focus on classifying the seven most abundant VF types and report superior performance compared to existing sequence- or structure-only methods, demonstrating the synergistic benefit of combining both data modalities. While the work is promising and the methodology is sophisticated, several points should be addressed to elevate the manuscript to the standard of a top-tier journal.

1. A critical missing element is the evaluation of the quality of the predicted 3D structures from ESMFold. The entire structure-based channel relies on these predictions. The authors should report the predicted Local Distance Difference Test (pLDDT) scores for the dataset. It is crucial to analyze if there is a correlation between low-confidence structure predictions and model misclassifications. A model trained on inaccurate or low-confidence structures may learn artifacts rather than true biological signals.

2. The manuscript states that VFs longer than 1240 amino acids were excluded due to GPU memory limitations. This is a significant constraint that could introduce bias. The authors should provide a more precise quantification of what percentage of the total VFs in the initial dataset this exclusion represents. Furthermore, a discussion is needed on the potential biological implications, as longer proteins may possess complex domain architectures or functions that the model is now unable to learn.

3. The model's dataset is limited to the seven most numerous VF categories, excluding seven rarer ones. While this is a practical decision for model training, it limits the claimed generalizability of the model. The authors should be more explicit about this limitation and frame their conclusions accordingly, acknowledging that the model's utility for less common VF types is currently unknown.

4. The method for fusing the sequence and structure channel embeddings is a simple summation. This architectural choice requires justification. The authors should explore and compare this with other fusion strategies, such as concatenation followed by a linear layer or an attention-based fusion mechanism, to demonstrate that summation is indeed an optimal or sufficient choice.

5. The analysis of remote homology detection is compelling but appears anecdotal. To strengthen this claim, the authors should perform a systematic analysis. They could identify all pairs in the test set that qualify as remote homologs (e.g., sequence identity < 30%, TM-score > 0.5) and report the model's performance specifically on this subset, rather than presenting only two cherry-picked examples.

6. When comparing PLM-GNN with other models (Table 3), it is not specified whether the hyperparameters for these baseline models were also rigorously optimized. For a fair comparison, all competing models should be tuned to their best performance on the validation set, just as the proposed model was. The authors should clarify their procedure for this.

7. While the performance improvements of PLM-GNN are shown, the authors have not reported whether these improvements are statistically significant. Statistical tests (e.g., McNemar's test or bootstrapping to generate confidence intervals for the metrics) should be performed on the performance differences between PLM-GNN and the next-best-performing models to validate the superiority of the proposed method.

8. The paper would benefit from a more in-depth error analysis beyond the confusion matrix. The authors should investigate the characteristics of the misclassified VFs. For example, are they more likely to have low pLDDT scores, belong to specific organisms, have lengths close to the 1240 residue cutoff, or share features with another class that explains the confusion?

9. The description of the geometric features for the graph representation (Table 2) is very dense. While comprehensive, its complexity could hinder reproducibility. A more detailed, step-by-step explanation in the supplementary materials, perhaps with a visual diagram illustrating how these features are derived from a protein structure, would greatly improve clarity.

10. The conclusion that the model "can provide valuable theoretical support for the development of antiviral strategies" is an overstatement. The model is a classification tool, which is an important step, but it does not directly provide mechanistic insights or drug targets. A more measured and precise conclusion regarding the model's immediate application—improving the functional annotation of proteins—would be more appropriate.

11. In the "Applied to effector delivery system classification" section, the model's performance on the T7SS category is noted to be weaker due to the small sample size. This highlights a potential weakness in handling highly imbalanced data. The authors could discuss or experiment with advanced data augmentation techniques or few-shot learning approaches as potential future work to address this.

12. The interpretability of the model is primarily explored through t-SNE visualizations. To further this, the authors could consider using model interpretability techniques like integrated gradients or attention map analysis, especially on the Transformer components, to identify which specific amino acid residues or structural motifs are most influential for classifying a given VF type. This would provide more direct biological insights.

13. For a more comprehensive comparison, further studies involving standalone Convolutional Neural Network (CNN) and Vision Transformer (ViT) models could be included. Comparing the proposed method against other sophisticated, single-modality architectures would better highlight the specific advantages of the dual-channel approach. "Hybrid deep learning model for automated colorectal cancer detection using local and global feature extraction","InceptionNeXt-Transformer: A novel multi-scale deep feature learning architecture for multimodal breast cancer diagnosis"

Reviewer #2: Reviewer Report

This paper introduces PLM-GNN, a dual-channel model for accurate classification of seven major virulence factors. By integrating structural and sequence-based features, the model achieves high performance, aiding in the study of pathogen virulence mechanisms. While the core contributions appear sound, the presentation lacks critical clarifications in methodology and notation that impact reproducibility and readability. I recommend a major revision, as the following methodological clarifications are essential for clarity and rigor.

Major Comments

1. The author should add the justification for the 8:1:1 stratified split

The paper states:

“We then randomly divided the data into three subsets using an 8:1:1 ratio (with stratified sampling).”

2. The manuscript relies heavily on undefined acronyms (e.g., “CNN”, “SVM”, “Cα”). Please define each acronym at its first occurrence.

3. Several symbols appear without explanation, like (X(0)∈RL×1280X^{(0)} )

For each symbol, please include a brief textual definition at the point of first use.

**Do you want your identity to be public for this peer review?** For information about this choice, including consent withdrawal, please see our Privacy Policy

Reviewer #1: No

Reviewer #2: No

---

## [Author Response · Author response to Decision Letter 1]

5 Sep 2025

Dear Editor,

On behalf of my co-authors, we thank you very much for giving us an opportunity to revise our manuscript. We are truly grateful to the comments and suggestions from you and the reviewers. Those comments are very helpful for revising and improving our paper. We have studied comments carefully and tried our best to revise and improve the manuscript and made great changes in the manuscript according to the reviewers’ comments. Revised portions are marked in yellow in the paper. The main corrections in the paper and the responses to the reviewer’s comments are listed as following:

Reviewer #1

1. Comment: A critical missing element is the evaluation of the quality of the predicted 3D structures from ESMFold. The entire structure-based channel relies on these predictions. The authors should report the predicted Local Distance Difference Test (pLDDT) scores for the dataset. It is crucial to analyze if there is a correlation between low-confidence structure predictions and model misclassifications. A model trained on inaccurate or low-confidence structures may learn artifacts rather than true biological signals.

Response: We thank the reviewer for raising this critical point regarding the quality of ESMFold-predicted structures and its relationship with the reliability of our structure-based predictions. We fully agree that assessing prediction confidence is essential for interpreting model outputs. In response to this comment, we have now included a comprehensive evaluation of the pLDDT scores across our dataset and explicitly examined their correlation with model performance.

We computed the mean pLDDT per protein as an overall confidence metric. As shown in S1 File’s S1 Fig, the global distribution of pLDDT values is right-skewed, with a mean of 76.46. The majority of residues fall into medium- to high-confidence intervals (70–90: 54.3%; >90: 18.9%), though a proportion of low-confidence predictions (<50) remains (10.0%). We also observed certain variations across categories; for instance, “Biofilm” and “Nutritional/Metabolic factor” categories showed high mean confidence (84.7 and 84.9, respectively), while “Effector delivery system” had lower confidence (mean = 70.1).

Most importantly, we directly assessed whether low-confidence predictions correlate with misclassifications by our model (PLM-GNN). We computed both Pearson and Spearman correlation coefficients between the mean pLDDT scores and model classification errors. The results show no significant correlation (Pearson p = 0.1519; Spearman p = 0.0987), indicating that our model exhibits strong robustness. We hypothesize that this may be attributed to the overall high pLDDT values in the dataset, with a small number of low-pLDDT samples having minimal impact on the overall model performance.

These analyses have been added to the S1 File “Distribution of pLDDT and Correlation with Model Performance”. We acknowledge that some regions exhibit low confidence and appreciate the reviewer’s emphasis on careful interpretation. Still, the absence of a significant correlation suggests that our model does not systematically rely on low-quality structural artifacts for classification.

Thank you again for this insightful suggestion, which has strengthened our evaluation.

2. Comment: The manuscript states that VFs longer than 1240 amino acids were excluded due to GPU memory limitations. This is a significant constraint that could introduce bias. The authors should provide a more precise quantification of what percentage of the total VFs in the initial dataset this exclusion represents. Furthermore, a discussion is needed on the potential biological implications, as longer proteins may possess complex domain architectures or functions that the model is now unable to learn.

Response: We thank the reviewer for this insightful comment regarding the exclusion of longer VFs and its potential implications. We agree that this filtering step may introduce bias, and we have now provided further clarification and quantification in the revised manuscript.

Specifically, the initial dataset contained 4,020 VF sequences. After removing sequences with >90% homology using CD-HIT, 3,880 sequences remained. Among these, 128 sequences (3.29% of the dataset) were longer than 1240 amino acids and were excluded due to GPU memory constraints. We have explicitly stated these proportions in the Methods section’s subsection “Dataset description” (lines 167-172) for clarity.

We fully acknowledge that the exclusion of longer proteins may limit the model's ability to learn from complex, multi-domain VFs, which are often associated with sophisticated mechanisms. To better contextualize this limitation, we included an analysis of the length distribution of VFs across categories (now in Fig 2). This analysis confirms that long sequences are rare and non-uniformly distributed among classes.

Nevertheless, given that such long sequences constitute a very small fraction (3.29%) of the overall dataset, we expect that their exclusion has not substantially skewed the model’s general representativeness or performance. In the Discussion and conclusion section(lines 659–663), we have further reflected on this limitation and stated plans for future work to overcome computational constraints—such as employing larger-memory GPUs or developing a segmenting strategy—to include all VFs regardless of length.

Thank you again for raising this important point, which has helped us improve the clarity and rigor of our manuscript.

3. Comment: The model's dataset is limited to the seven most numerous VF categories, excluding seven rarer ones. While this is a practical decision for model training, it limits the claimed generalizability of the model. The authors should be more explicit about this limitation and frame their conclusions accordingly, acknowledging that the model's utility for less common VF types is currently unknown.

Response: We thank the reviewer for this valuable comment. We fully agree that the restriction of our training dataset to the seven most abundant VF categories represents a significant limitation to the claimed generalizability of PLM-GNN.

In the revised manuscript, we have explicitly stated in the Discussion and conclusion section (lines 652–656) that our model was trained and evaluated only on the majority categories and that its performance on the seven excluded, rarer VF categories remains entirely unknown. This limitation unavoidably constrains the model's applicability and predictive power for under-represented VF types. We have therefore reframed our conclusions to more cautiously reflect that the current model is validated primarily for common VF categories, and any broader claims about generalizability across all VF types would be premature.

To address this in future work, we plan to actively incorporate samples from these rare categories, possibly employing few-shot learning or data augmentation techniques, to build a more comprehensive and universally applicable model. We appreciate the reviewer’s emphasis on this point, which helps provide a more accurate interpretation of our results.

4. Comment: The method for fusing the sequence and structure channel embeddings is a simple summation. This architectural choice requires justification. The authors should explore and compare this with other fusion strategies, such as concatenation followed by a linear layer or an attention-based fusion mechanism, to demonstrate that summation is indeed an optimal or sufficient choice.

Response: We thank the reviewer for raising this important question regarding our feature fusion strategy. We fully agree that the choice of fusion method is a critical architectural decision that requires thorough justification.

In response to this comment, we systematically explored and compared three distinct fusion strategies: (1) direct addition, (2) concatenation, and (3) adaptive weighted fusion based on an attention mechanism. As shown in S3 File’s S4 Fig of the Supplementary Information, our comparative analysis demonstrated that the direct addition strategy achieved the best performance on the validation set for the VF classification task. Although we initially anticipated that more complex mechanisms such as attention might outperform simpler methods, the additive fusion consistently delivered superior results in our experiments. We hypothesize that this may be because the feature embeddings from the sequence and structure channels are already well-aligned in the same latent space, allowing their direct summation to serve as an effective and efficient means of feature integration.

We have now expanded the description of the fusion strategy comparison in the Methods section’s subsection“Classifying VFs and weighted cross-entropy loss”(lines 328-339) to make the rationale for this choice more explicit. We also include a brief discussion of the experimental results—noting that while additive fusion proved optimal for our specific model and dataset, the best strategy may vary for other tasks, and the choice of fusion method should always be validated empirically.

Thank you for prompting us to clarify this essential aspect of our model architecture.

5. Comment: The analysis of remote homology detection is compelling but appears anecdotal. To strengthen this claim, the authors should perform a systematic analysis. They could identify all pairs in the test set that qualify as remote homologs (e.g., sequence identity < 30%, TM-score > 0.5) and report the model's performance specifically on this subset, rather than presenting only two cherry-picked examples.

Response: We thank the reviewer for this insightful suggestion. We agree that a more systematic evaluation is essential to robustly demonstrate the model's capability in detecting remote homology.

In response to this comment, we have now conducted a comprehensive analysis to quantitatively evaluate the model’s performance on remote homologs. We systematically identified all protein pairs between the training and test sets that meet the stringent criteria for remote homology (sequence identity < 30% and TM-score > 0.5). This process resulted in a dedicated evaluation subset comprising 93 test samples, all of which are remote homologs of sequences in the training set. PLM-GNN achieved an accuracy of 91.4% on this challenging subset (85 out of 93 samples predicted correctly). This performance significantly surpasses that of the sequence channel (79 out of 93 correct) and the structure channel (70 out of 93 correct), highlighting the critical importance of integrating both sequence and structural information for recognizing distantly related analogs. We have added this systematic analysis to the Results section’s subsection “Accurately classifying VFs with remote homology”(lines 623-635) and incorporated the quantitative results into the revised manuscript.

Thank you for this valuable suggestion, which has greatly enhanced the persuasiveness of our study.

6. Comment: When comparing PLM-GNN with other models (Table 3), it is not specified whether the hyperparameters for these baseline models were also rigorously optimized. For a fair comparison, all competing models should be tuned to their best performance on the validation set, just as the proposed model was. The authors should clarify their procedure for this.

Response: We thank the reviewer for their important comments regarding the hyperparameter optimization of the baseline models, which are essential for ensuring a fair comparison.

We have re-optimized the parameters for all baseline models and evaluated their performance on the test set using the hyperparameter configurations that achieved the best performance on the validation set. All baseline models listed in S3 File’s Table S1 of the supplementary information underwent this tuning process. For each type of baseline method, we conducted a systematic grid search within the hyperparameter ranges provided in S3 File’s Table S1, covering key architectural and training parameters such as the number of attention heads (for attention-based models), hidden layer dimensionality, learning rate, and number of layers. Each model configuration was evaluated on the same validation set used for PLM‑GNN, and the best-performing configuration was selected for the final comparison in Table 3. Additionally, Table 3 and Fig 6 in the main text have been updated accordingly.

This systematic optimization process ensures that all models achieve their best possible performance on the dataset used in this study, thereby enabling a fair and meaningful comparison.

We appreciate the reviewer’s suggestion to clarify this aspect, which further enhances the credibility of our comparative analysis.

7. Comment: While the performance improvements of PLM-GNN are shown, the authors have not reported whether these improvements are statistically significant. Statistical tests (e.g., McNemar's test or bootstrapping to generate confidence intervals for the metrics) should be performed on the performance differences between PLM-GNN and the next-best-performing models to validate the superiority of the proposed method.

Response: We thank the reviewer for this insightful suggestion regarding the statistical validation of our model's performance. We agree that demonstrating the statistical significance of the improvements is essential.

In response to this comment, we have now performed a comprehensive statistical significance analysis to compare PLM-GNN against the top-performing baseline models. The test set was divided into five equally sized subsets. In each iteration, four subsets were used for evaluation, resulting in five independent performance measurements for each model. We then conducted independent samples t-tests between PLM-GNN and each of the best-performing baselines (GCN and GAT) across multiple evaluation metrics.

As shown in Fig 5, PLM-GNN demonstrated consistent and statistically significant superiority over both comparison models, with all p-values being less than 0.05. This analysis confirms that the performance improvements achieved by our model are not only consistent but also statistically significant. We have added these results to the manuscript in the Methods section’s subsection " Comparison with other methods in terms of VF classification"(page 23) to provide rigorous statistical validation of our conclusions.

8. Comment: The paper would benefit from a more in-depth error analysis beyond the confusion matrix. The authors should investigate the characteristics of the misclassified VFs. For example, are they more likely to have low pLDDT scores, belong to specific organisms, have lengths close to the 1240 residue cutoff, or share features with another class that explains the confusion?

Response: We thank the reviewer for this insightful suggestion regarding a more comprehensive error analysis. We have conducted an in-depth investigation into the characteristics of misclassified VFs, focusing specifically on the potential influencing factors mentioned in the comment.

First, in the S1 File“Distribution of pLDDT and Correlation with Model Performance,”we computed both Pearson and Spearman correlation coefficients between the misprediction label vector of the PLM-GNN and the pLDDT score vector. The results indicated no significant correlation, suggesting that model errors are not attributable to low-confidence structure predictions. Next, we examined whether misclassifications were concentrated in specific organisms. As shown in S3 File’s S2 Fig (Error rate heatmap of bacterial genera by VF category), no particular genus exhibited significantly elevated error rates, indicating that taxonomic origin is not systematically associated with misclassification. We then analyzed whether sequence length near the 1,240-residue cutoff influenced errors. Sequences were grouped into intervals (each spanning 200 residues), and performance metrics were calculated for each group. S3 File’s S3 Fig demonstrates that sequences in longer intervals (1000-1,240 residues) performed comparably to those in shorter intervals (200-400 residues), suggesting no clear relationship between sequence length and misclassification. Finally, t-SNE visualization of the fused features (Fig 7D) revealed that samples from differe

---

## [Decision Letter · Decision Letter 1]

18 Dec 2025

Classification of virulence factors based on dual-channel neural networks with pre-trained language models

PONE-D-25-34906R1

Dear Dr. Li,

We’re pleased to inform you that your manuscript has been judged scientifically suitable for publication and will be formally accepted for publication once it meets all outstanding technical requirements.

Kind regards,

Additional Editor Comments (optional):

Reviewers' comments:

Reviewer's Responses to Questions

**Comments to the Author**

Reviewer #1: All comments have been addressed

Reviewer #3: All comments have been addressed

2. Is the manuscript technically sound, and do the data support the conclusions?

Reviewer #1: Yes

Reviewer #3: Yes

3. Has the statistical analysis been performed appropriately and rigorously?

Reviewer #1: No

Reviewer #3: Yes

4. Have the authors made all data underlying the findings in their manuscript fully available?

Reviewer #1: No

Reviewer #3: Yes

5. Is the manuscript presented in an intelligible fashion and written in standard English?

Reviewer #1: Yes

Reviewer #3: Yes

Reviewer #1: Accept , comments have been adressed. Paper can be accepted as its current form

Accept , comments have been adressed. Paper can be accepted as its current form.

Reviewer #3: In this paper, the authors propose a deep network model to classify virulence factors. The model is called PLM-GNN that includes CNN and transformer structures.

The proposed approach can be helpful for other researchers. However, the following revisions are required to enrich and improve the quality of the paper;

1) It is commonly known that traditional machine learning-based methods are not efficient, and deep learning-based methods produce better results.

Therefore, the statements about thraditional machine learning (e.g., SVM) in the introduction should be removed. For instance, the sentences "Garg et al. [12] introduced VirulentPred, a prediction .....with machine learning models, which yielded promising results."

They are redundant/unnecessary.

2) The meaning of the statement ".....cross-entropy loss is widely adopted as the standard loss function" should be supported by the following works: https://doi.org/10.1016/j.bspc.2025.108083,
https://doi.org/10.1016/j.bspc.2025.108138,
https://doi.org/10.1016/j.eswa.2024.126290

3) In this work, RELU has been used for activations. However, it may cause dying neuron issues. Therefore, leaky RELU has been used in different network models to obtain higher performance.

This can be considered a limitation of this work. In future work, the proposed PLM-GNN structure can be implemented using leaky ReLU to improve its performance.

To inform the readers (who may want to apply the same model) about this, the following statements should be in the last paragraph:

"A potential future work can be applying the proposed PLM-GNN structure using leaky ReLU rather than ReLU and investigating its performance because leaky ReLU has been used in various recent models [https://doi.org/10.1016/j.neucom.2024.127445, https://doi.org/10.1007/s11042-025-20760-y, https://doi.org/10.1007/s10462-024-10897-x, https://doi.org/10.1016/j.bspc.2025.108370]."

**Do you want your identity to be public for this peer review?** For information about this choice, including consent withdrawal, please see our Privacy Policy

Reviewer #1: No

Reviewer #3: No

---

## [Editor Report · Acceptance letter]

PONE-D-25-34906R1

PLOS One

Dear Dr. Li,

I'm pleased to inform you that your manuscript has been deemed suitable for publication in PLOS One. Congratulations! Your manuscript is now being handed over to our production team.

Kind regards,

on behalf of

Dr. Aiqing Fang

Academic Editor

PLOS One